# Effects of Chromium-L-Methionine in Combination with a Zinc Amino Acid Complex or Selenomethionine on Growth Performance, Intestinal Morphology, and Antioxidative Enzymes in Red Tilapia *Oreochromis* spp.

**DOI:** 10.3390/ani12172182

**Published:** 2022-08-25

**Authors:** Rawiwan Limwachirakhom, Supawit Triwutanon, Srinoy Chumkam, Orapint Jintasataporn

**Affiliations:** 1Department of Aquaculture, Faculty of Fisheries, Kasetsart University, Bangkok 10900, Thailand; 2Faculty of Veterinary Medicine, Kasetsart University, Kamphaeng Saen, Nakhon Pathom 73140, Thailand; 3Faculty of Agricultural Technology, Valaya Alongkorn Rajabhat University, Pathum Thani 13180, Thailand

**Keywords:** chromium-L-methionine, zinc amino acid complex, selenomethionine, organic trace mineral, red tilapia, growth performance, feed utilization, immune response

## Abstract

**Simple Summary:**

The purpose of this study was to develop diet optimization for the growth and health of fish under intensive aquaculture systems, with a focus on the farming of Nile tilapia and red tilapia in Thailand. Diets and proper nutrition are more important than ever in raising healthy, antibiotic-free fish. Organic trace minerals have been used to promote the growth and health status of fish in aquafeeds. In this investigation, we studied the effects of chromium-L-methionine, in combination with a zinc amino acid complexor in combination with selenomethionine, on the growth performance, feed utilization, and immune responses of red tilapia. The results for the combination of chromium-L-methionine with a zinc amino acid complex showed beneficial effects for red tilapia in terms of growth performance, feed utilization, and intestinal morphology. The results for the combination of chromium-L-methionine with selenomethionine demonstrated that the levels of antioxidative enzymes, especially glutathione, could be increased to provide an innate immune response in red tilapia.

**Abstract:**

To consider diet optimization for the growth and health of fish under intensive aquaculture systems, with a focus on the farming of Nile tilapia and red tilapia in Thailand, we conducted an experiment based on a completely randomized design (CRD), with three treatments and four replicates. Three diets, supplemented with different trace minerals, were applied to selected groups of fish: (a) a control diet, without organic trace minerals supplementation; (b) a T1 diet of chromium-L-methionine at 500 ppb, in combination with a zinc amino acid complex at 60 parts per million (ppm); and (c) a T2 diet of chromium-L-methionine at 500 ppb in combination with selenomethionine at 300 ppb. Red tilapia with an initial mean weight of 190 ± 12 g/fish were randomly distributed into cages of 2 × 2 × 2.5 m in a freshwater pond (12 cages in total), with 34 fish per cage and a density of 17 fish/m^3^. During the 8 week feeding trial, the fish were fed 3–4% of their body weight twice a day. The fish were weighed, then blood samples were collected to study their immune responses. The intestines were collected, measured, and analyzed at the end of the feeding trial. The results showed that the red tilapia that were fed with diets of chromium-L-methionine in combination with a zinc amino acid complex in the T1 treatment had significantly (*p* < 0.05) higher final weights, weight gains, average daily gains (ADGs), and better feed conversion ratios (FCRs), compared with fish that were fed with the control diet without organic trace minerals and with fish that were fed with the T2 diet (*p* < 0.05). The midgut and hindgut villus heights of the group fed with chromium-L-methionine in combination with a zinc amino acid complex in the T1 treatment were significantly higher than those of the other groups (*p* < 0.05). The levels of the antioxidative enzyme superoxide dismutase (SOD) and lysozyme activity were not significantly different from those of fish that were fed with the control diet (*p* > 0.05), whereas the glutathione level tended to increase (*p* < 0.1) in fish that were fed with chromium-L-methionine in combination with selenomethionine in the T2 treatment. Therefore, we concluded that chromium-L-methionine in combination with a zinc amino acid complex or selenomethionine clearly enhanced red tilapia’s growth performance and feed utilization through the promotion of antioxidative enzyme activity and immune response.

## 1. Introduction

Nile tilapia and red tilapia (*Oreochromis* spp.) are economically important freshwater fish in Thailand. Red tilapia, in particular, is ranked second in freshwater fish production. It is a high-value export to other countries [1]. In addition, the color of red tilapia indicates a premium marine species and provides significant market appeal [2], as do its firm texture, rapid growth, salinity tolerance, and adaptability to most cultural systems [3]. However, the production costs of red tilapia vary widely, due to climate and environmental conditions that result in sudden deaths or contagious diseases in the fish populations. 

Optimal nutrition is an essential factor in maintaining and enhancing immunity under conditions of stress during aquaculture, so that quality customer products may be provided while maintaining genetic diversity under the current conditions of climate change [4]. Commercial aquafeeds often contain lower concentrations of trace minerals that are caused by the replacement of fishmeal with alternatives, such as terrestrial plant-based proteins [5]. Soybean meal, corn, rice, and wheat are added to commercial fishmeal aquafeeds. However, plant-based proteins, especially soybean meal, contain phytic acid, which inhibits trace mineral absorption as an antinutritional factor. Phytic acid inhibits the absorption of zinc, calcium, magnesium, copper, selenium, iron, cobalt, and nickel by forming very stable complexes, thereby reducing their bioavailability [6]. Replacing fishmeal with plant-based proteins limits the amount and the availability of essential trace minerals, such as zinc and selenium, resulting in reduced fish growth, poor health, and the excretion of unabsorbed trace minerals that contaminate the environment [7]. The extensive biochemical functions of trace minerals are required by more than one-third of all proteins for metabolic activities, cell replication, antioxidant activity, and immune responses. However, red tilapia’s requirements for the optimal levels or combinations of essential trace minerals to develop nutritional diets and immunities, under current aquaculture production systems, is unknown [8].

Trivalent chromium is one trace mineral that can stimulate carbohydrate and lipid metabolism, by increasing the insulin receptor levels that maximize growth performance [9]. In addition, chromium binds to oligopeptide and forms chromodulin; then, chromodulin binds with activated substances to activate insulin receptors before enhancing insulin’s functions [10]. Many studies have demonstrated chromium’s ability to increase insulin activity and promote lipid metabolism; accordingly, chromium is an essential trace mineral for animals [11,12]. 

In tilapia, diets with chromium supplementation of 400 ppb to 600 ppb can enhance physiological and biochemical conditions [13]. Dietary chromium at the level of 400 ppb has been reported to increase the crude protein content of tilapia carcasses and the gonadosomatic index (GSI) of Nile tilapia fingerlings [14]. Some studies of chromium yeast have shown the benefits of modulating the immune responses of rainbow trout [15].

Most of the research into the biochemical functions of zinc has focused on the identification of the biological benefits for animals. in terms of catalysis, structure, and regulatory systems. The catalytic function of zinc is essential for biochemical mechanisms that are related to cofactors of more than 300 enzymes, e.g., alkaline phosphatase, carbonic anhydrases, and RNA nucleotide transferase [16]. In addition, zinc is an essential trace mineral, corresponding to metalloenzymes such as Cu-Zn superoxide dismutase. The functional and structural integrity of several transcription factors, metabolic pathways, and signaling pathways are based on the benefits of zinc-requiring proteins [17]. For example, metallothionein is a zinc-binding protein that can activate or inhibit gene expression regulation. The chelation of dietary zinc with amino acid (histidine or cysteine) may promote zinc absorption and distribution throughout the tissue of fish [18]. There have been several reports on the zinc requirements in diets that are related to growth, feed efficiency, whole body zinc concentration and retention, plasma or serum levels, and enzyme activities—especially for juvenile fish species. The zinc supplementation requirement in several fish is 33.5 ppm to 64.6 ppm [19].

Oxidative damage is the result of the release of toxic free radicals, such as hydrogen peroxide (H_2_O_2_), from metabolic pathways, and of stress in animals, which negatively affect health and disease prevention [20,21,22,23]. Hydrogen peroxide signaling and detoxification are decreased by the glutathione peroxidases that are involved in the maintenance of cellular redox homeostasis. Selenium is an important micronutrient that is a composite of selenoproteins. It manages special biological functions, especially with respect to selenocysteines [24]. Selenocysteines are part of the active sites of antioxidative enzymes, such as glutathione peroxidases, that can reduce the release of hydrogen peroxide or free radicals from lipid peroxidation, such as hydroxyl (OH^−^) or hydroperoxyl (HOO^−^), while oxidizing glutathione [25]. Glutathione (GSH) is a major free thiol that is present inside cells, mainly in a reduced form (90% to 95% of total glutathione, GSH + GSSG). The oxidation of glutathione leads to the formation of glutathione disulfide (GSSG) in redox reactions [26]. GSSG occurs chemically, but it is also catalyzed by enzymes that are able to use GSH to reduce H_2_O_2_ or other peroxides into water or into corresponding alcohol. In terms of glutathione metabolites, glutathione S-transferase (GST) has the important ability to catalyze the conjugation of the reduced form of glutathione to xenobiotic substrates for the purpose of detoxification. Xenobiotic substrates that are released in response to chemical stress have been highlighted as impactful environmental contaminants that generate increasing concern, due to the expansion of human population and the generation of deleterious effects in aquatic species due to persistent organic pollutants, pesticides, toxins, metals, pharmaceuticals, nanomaterials, and temperature or oxygen changes in water [27]. 

Moreover, selenium is the constituent of thioredoxin reductase that maintains cellular redox status and protects cells from oxidative damage. Iodothyronine deiodinase is, another selenoprotein that can activate nutrient metabolism by catalyzing the inactivation of thyroxine (T4) to diiodothyronine (T3) [24]. Selenium supplementation can enhance the expression of selenoproteins in zebrafish and rainbow trout [28]. 

In channel catfish, the requirements of selenium in diets are based on weight gain, whole body selenium retention, and glutathione peroxidase activities in the liver [29]. The organic forms of selenium, especially selenomethionine and selenocysteine, are more bioavailable than inorganic selenium forms, such as selenate, selenite, and selenide [24,29]. The minimum selenium requirements of several fish are based on the form of selenium (inorganic or organic), the feed ingredients, the amount of vitamin E, and the concentrations of waterborne selenium. In channel catfish, based on weight gain and glutathione peroxidase activities, the amount of selenium supplementation required to improve growth performance and antioxidative enzymes is estimated to be 90 ppb to 120 ppb of organic selenium forms [19,29] and 170 ppb to 280 ppb of inorganic forms, such as sodium selenate [30]. In addition, dietary supplementation with 570 ppb of selenomethionine can promote weight gain and glutathione peroxidase activity in the liver of Nile tilapia [19].

Currently, organic trace minerals are employed in aquafeeds to promote absorption rates that are higher than those of inorganic trace minerals, with more potential to absorb into the small intestine [31,32]. Moreover, chelated minerals or organic trace minerals have stable structures and are less sensitive or conjugated to phytate, which is a seriously antinutritional factor in plant-based diets [33,34]. However, research into trace minerals or the optimal requirements for grower feed for fish, such as Nile tilapia (hybrid of *O. niloticus × O. aureus*) and red tilapia, has rarely been aimed at identifying dose recommendations or the necessity of using trace mineral supplements. The supplementation of organic trace minerals, such as zinc in combination with chromium, has the potential to promote growth, nutrient metabolism, and immune response; using chromium in combination with selenium has the potential to enhance growth and antioxidative enzymes. Therefore, the aim of this study was to evaluate the effects of supplementations of chromium-L-methionine in combination with a zinc amino complex or chromium-L-methionine in combination with selenomethionine on the growth performance and immunity of red tilapia.

## 2. Materials and Methods

### 2.1. Experimental Design

The experiment was a completely randomized design (CRD) study with three treatments and four replications: (a) a control diet without organic trace mineral supplementation, (b) a T1 diet of chromium-L-methionine at 500 ppb in combination with a zinc amino acid complex at 60 ppm, and (c) a T2 diet of chromium-L-methionine at 500 ppb in combination with selenomethionine at 300 ppb.

### 2.2. Diets Description

The formulations of the three trial diets with the inclusion of three chelated minerals—chromium-L-methionine (organic trace mineral form, USA), a zinc amino acid complex (organic trace mineral form, USA), and selenomethionine (organic trace mineral form, USA)—are shown in Table 1. The basal diet comprised 32.51% crude protein, 5.00% crude lipid, 7.77% moisture, 2.79% fiber, 9.00% ash, 1.63% calcium, 1.15% phosphorus, and 4440.24 kcal/kg of energy, which was sufficient for optimal growth [35]; the concentrations of chromium, zinc, and selenium were 1440 ppb, 117.60 ppm, and 504 ppb, respectively. Fishmeal, soybean meal, corn gluten meal, micronutrient preservation premix, and other ingredients were ground with a hammer mill, mixed until homogeneous, and then mixed with three chelated minerals. Water was added to the mixed feeds at a proportion of 25% to make the paste, which was then pelletized with a farm extruder and dried at 80 °C for 8 h until the moisture content was lower than 10%. The diets were stored in plastic bags at room temperature until used.

### 2.3. Experimental Condition

The red tilapia were obtained from a commercial tilapia farm, Ayutthaya province, Thailand, and were acclimatized for 10 days before grading to the same size and being randomly stocked in cages. with an initial mean weight of 190 ± 12 g/fish at a density of 34 fish (17 fish/m^3^) per cage. Each cage served as a replicate. During the 8 week feeding trial, the water quality was managed in optimum conditions for fish culture: salinity of 0–0.2 ppt, pH of 6.5–8.5, dissolved oxygen (DO) content of more than 4 mg/L, alkalinity of 70–120 mg/L, and hardness of 80–150 mg/l. The fish were fed the trial diet to apparent satiation 2 times a day at 09.00 am and 4.00 pm, at around 3% to 4% of their body weight, for 8 weeks to gain a weight of around 600 g. The growth performance of the fish was evaluated by randomly weighing four fish from each replicate every 2 weeks and adjusting the amount of feed provided. The amount of feed consumed for every experimental unit was recorded every 2 weeks. At the termination of the experiment, the fish were fasted for 24 h before harvest, and all fish were weighed and counted.

### 2.4. Data Collection

#### 2.4.1. Growth Performance

Based on the measurements of fish weight and feed consumption, the weight gain, the average daily weight gain, the specific growth rate, and the feed conversion ratio were evaluated, using the following equations:Weight gain (g/fish) = average final weight − average initial weight
Average daily weight gain (ADG; g/fish/d) = (average final body weight − average initial body weight)/number of days
Specific growth rate (SGR; %/day) = (Ln (final weight) − Ln (initial weight)) × 100/number of days
FCR = feed given (dry weight)/weight gain (wet gain)

#### 2.4.2. Measurement of Intestinal Villus Height

After 8 weeks of feeding, three areas of the intestines were collected and immediately dissected with sterile scissors (*n* = 4). The intestines were removed from fish abdominal cavities and dissected in three portions: the anterior part (from after the pyloric part of the stomach to before the spiral part of the intestine), the middle part (the spiral part of the intestine), and the posterior part (from after the spiral part of the intestine to 2 cm before the anus) [36]. The parts were separately collected and fixed in 10% buffer formalin. Fixed tissues were prepared and processed according to standard histological techniques. The tissues were dehydrated in ethanol and cleared with xylene before being embedded in paraffin. Tissue sections were transversally cut at 6 μm and then mounted on glass slides. After fixing the tissues, the sections were deparaffinized with xylene and stained according to the hematoxylin and eosin (H&E) method [37]. To measure villus height, each intestinal villus was photographed and measured from the villus tip to the bottom line. The lengths of the villi of each treatment were evaluated in terms of the average mean of their height per portion [36].

#### 2.4.3. Hematological Parameters and Antioxidative Enzymes

After 8 weeks of the experiment, blood (1 mL for whole blood and 2 mL for serum) was withdrawn from two fish randomly per replicate of each treatment. The fish were removed from cages and then anesthetized with clove oil (100 ppm). The whole blood collection comprised a punctured flow from the caudal vein using an EDTA solution as the anticoagulant agent. The hematocrit was determined by using microhematocrit heparinized capillary tubes, a microhematocrit centrifuge, and a reader, and its values were expressed in terms of the percentage of blood cell volume [38]. The hemoglobin concentration was measured with the dyeing method of Drabkin and Austin [39]. The total erythrocyte/leucocyte content was counted with a Neubauer cell counting chamber or hemocytometer per 1 mm^3^ of blood. Two milliliters of serum were collected without an anticoagulant and centrifuged at 3000 rpm for 15 min. The supernatant was collected and stored at −20 °C. This supernatant was used to determine serum protein and immunoglobulin M contents, according to the method of Lowry et al. [40], and the amount of protein was expressed as grams per deciliter. Serum lysozyme levels were determined with an enzymatic assay of lysozyme products [41]. The spectrophotometric determination (A450, light path = 1 cm) was based on the lysis reaction from lysozyme activity. The unit definition was as follows: one unit of lysozyme activity was the activity required to produce a ΔA450 of 0.001 per minute at pH 6.24 and 25 °C using a suspension of *Micrococcus lysodeikticus* (Sigma-Aldrich, St. Louis, MO, USA) as the substrate in a 2.6 mL reaction mixture.

The activity of the antioxidative enzyme serum superoxide dismutase (SOD) was measured with an SOD Determination Kit (Sigma-Aldrich, Buchs, Switzerland), which allows for the assaying of SOD by utilizing a highly water-soluble tetrazolium salt, such as WST-1 (2-(4-Iodophenyl)-3-(4-nitrophenyl)-5-(2,4-disulfophenyl)-2H-tetrazolium, monosodium salt), that produces a water-soluble formazan dye upon reduction with a superoxide anion [42]. The rate of the reduction with O_2_ is linearly related to the xanthine oxidase (XO) activity and is inhibited by SOD. Therefore, the IC50 (50% inhibition activity of SOD or SOD-like materials) could be determined with a colorimetric method. Furthermore, since the absorbance of WST-1 formazan at 440 nm is proportional to the amount of superoxide anions, the SOD activity (as an inhibition activity) was quantified by measuring the decrease in color development at 440 nm. We used a commercial kit (Sigma-Aldrich, St. Louis, MO, USA) to measure the levels of total glutathione (GSSG + GSH) in the sera of the biological samples [43].

### 2.5. Statistical Analysis

All data were subjected to one-way ANOVA (analysis of variance) using statistical software. Differences between the means were tested by Tukey HSD for variances. Overall significance was defined as a level = 0.05, and the results were presented as mean ± SD (standard deviation). Alphabetical notation was used to identify significant treatment differences at *p* < 0.05. All research was conducted at the Nutrition and Aquafeed Laboratory, Department of Aquaculture, Faculty of Fisheries, Kasetsart University, Bangkok, Thailand.

## 3. Results and Discussion

### 3.1. Growth Performance

The growth performance and feed utilization results for the red tilapia that were fed experimental diets in combination with organic minerals are shown in Table 2.

The results showed that the red tilapia fed with diets of chromium-L-methionine in combination with a zinc amino acid complex in the T1 treatment had significantly (*p* < 0.05) higher final weights, weight gains, average daily gains, and feed conversion ratio values than those in the T2 treatment, which showed higher growth performance values compared with the fish that were fed with the control diet, without organic trace minerals. However, the specific growth rate showed no significant differences between the treatment and control groups.

In this study, additional chromium-L-methionine at 500 ppb in combination with a zinc amino acid complex at 60 ppm in the red tilapia diet showed more potential to improve growth performance. The aim in using trace mineral supplementation is to increase profitability, growth performance, and feed conversion. In commercial aquaculture, raw materials and micronutrient preservation premixes usually provide trace mineral components, such as zinc (Zn) and selenium (Se), while chromium is provided by other raw material sources. In the control group, the basal diet was sufficient for optimal growth [35], with concentrations of chromium, zinc, and selenium at 1440 ppb, 117.60 ppm and 504 ppb, respectively. The Nutrition Research Council (NRC) recommends that the minimum requirement for tilapia is 60 ppm zinc for maximum weight gain and immunity from disease. However, the selenium and chromium requirements in tilapia diet have not yet been identified [35]. This study investigated organic zinc components combined with organic chromium dietary supplementation and evaluated optimal levels for red tilapia growth.

Zinc participates in the regulation of cell proliferation in several ways and is essential to enzyme systems that influence cell division and proliferation [44,45]. In comparisons of the supplementation of zinc amino acid complexes at 20, 60, and 120 ppm, the 60 ppm dose has presented the highest growth performance in red tilapia [46] and is the recommended level of minerals in tropical fish [35,47], including tilapia. It has been reported that juvenile *Labeo rohita* require 62.58 ppm of zinc in their diet to improve growth performance and increase alkaline phosphatase activity [48]. Similarly, the effect of dietary zinc supplementation at 60 ppm in juvenile tilapia (*O. niloticus* × *O. aureus*) has been shown to enhance growth performance and carbohydrate utilization [49]. 

Chromium is an essential trace element for animals [50] and plays important roles in the nutrition and physiological responses of fish [51], which potentiate the action of insulin. Chromium is vital for the normal metabolism of carbohydrates and lipids [52,53]. Kegley et al. [54] determined that supplementation using inorganic chromium was poor because of its low bioavailability, while organic chromium, such as chelated chromium, chromium picolinate, and chromium yeast, was more beneficial. A study of fingerling grass carp (*Ctenopharyngodon idellus*) fed with chromium picolinate at 800 ppb demonstrated optimal levels of growth and feed utilization, showing higher triglyceride and high-density lipoprotein cholesterol (HDL-C) concentrations when compared with other treatments [55]. Supplementation using chelated chromium at 500 ppb in rainbow trout increased blood glucose clearance but did not impact on growth performance [56]. Mehrim [13] found that supplementation of chromium picolinate at 400 ppb in diet was most appropriate for Nile tilapia (Oreochromis niloticus) fingerlings. Chromium-L-methionine supplementation in tilapia, especially red tilapia, has been little studied in comparison with studies of other animal species. In broiler chicken diets, supplementation with chromium-L-methionine at 800 ppb enhanced both growth performance and carcass traits in chickens that were reared under heat-stress conditions [57], while growing-finishing pigs supplemented with chromium-L-methionine at 200 ppb in combination with a zinc amino acid complex at 50 ppm showed improved growth performance, feed conversion ratio, meat quality, and carcass traits when compared with the control group [58]. In the current study, the addition of chromium-L-methionine at 500 ppb in combination with a zinc amino acid complex at 60 ppm in the diet of red tilapia enhanced growth performance with improved feed utilization (*p* < 0.05). However, the results of growth performance may depend on single organic supplementation, e.g., a zinc amino acid complex or chromium-L-methionine combined with a zinc amino acid complex.

### 3.2. Measurement of Villus Height

Three portions of the intestine of the red tilapia fed with different diets are shown in Figure 1, and the calculated results of intestinal villus heights are shown in Table 3. The villus heights of the intestines of red tilapia that were fed with a supplementation of a zinc amino acid complex at 60 ppm in combination with chromium-L-methionine at 500 ppm were shown to be significantly higher in the middle and posterior parts of the intestine, when compared with those of the control and T2 groups (*p* < 0.05). However, there were no significant differences in the intestinal villus heights between the T2 group (chromium-L-methionine at 500 ppm in combination with selenomethionine at 300 ppm) and the control group, except for of the differences in the anterior intestine that were significantly lower in the T2 group (*p* < 0.05). In addition, the highest absorption surface areas were observed in fish supplemented with 500 ppm of chromium-L-methionine in combination with 60 ppm of a zinc amino acid complex.

The small intestine is an important organ in the digestive tract. It is most involved in the nutrient absorption of animals. Absorption generally occurs due to the paracellular permeability of the intestinal epithelium after digestion [59]. Hence, increasing nutrient absorption can be observed through the relative index of villus height and small intestine characteristics, such as the numbers of epithelial cells and the depth of villous crypts [60].

Studies of omnivorous tilapia (*O. niloticus* × *O. aureus*) demonstrated their similarities in function and structure with mammals with the highest absorption and digestibility. Abundances of digestive enzymes have been identified in the anterior areas of the intestine, whereas the absorption and organization of villi were found to be decreased in the posterior portions [61,62]. The small intestine has a characteristic of nested loops in the peritoneal cavities. It is typically involved in the crucial nutrient absorption of hybrid tilapia. The observations of the longer and narrower villi with high density, the wider distal villi branching from the anterior to the middle portions, and the goblet cell counts of the small intestine are essential when evaluating absorption availability and improving feed utilization. These characteristics are good indicators of a healthy intestine in aquatic animals [63].

This study demonstrated the effects of the dietary supplementation of chromium-L-methionine at 500 ppm in combination with a zinc amino acid complex at 60 ppm in red tilapia, which resulted in significantly higher intestinal villus lengths in the midgut and hindgut, compared with those in the control diet (*p* < 0.05). Previously, dietary zinc supplementation was shown to increase intestinal villus heights and digestive enzyme activities, to improve intestinal barrier function, and to promote beneficial health in Nile tilapia fingerings [64,65].

However, in fish, the organic forms of trace minerals, such as a zinc amino acid complex, accelerate nutrient absorption and enhance zinc absorption through the intestinal membrane more than inorganic forms [33,66]. A study of juvenile shrimp (*Litopenaeus vannamei*) showed that supplementations with a zinc amino acid complex at 60 ppm were better able to increase bioavailability, growth performance, feed utilization, meat quality, antioxidant capacity, and immune response, when compared with supplementation with zinc sulfate (inorganic form) [67]. In addition, for Nile tilapia, the supplementation of organic zinc at 40 ppm enhanced growth, intestinal morphology, immune response, and antioxidative resistance to heat stress [64,68].

### 3.3. Hematological Parameters and Immune Response

Hematological parameters are basic blood test indicators that are closely associated with systemic metabolic status. Therefore, they can be used to effectively assess body health conditions. The effects on the hematological parameters and immune response of juvenile red tilapia of the supplementation of dietary chromium-L-methionine at 500 ppm in combination with a zinc amino acid complex at 60 ppm in the T1 treatment, and the supplementation of chromium-L-methionine at 500 ppm in combination with selenomethionine at 300 ppm in the T2 treatment, are presented in Table 4.

Under normal conditions, no significant differences were found in the hematological parameters (*p* > 0.05) of RBC, WBC, hematocrit, and hemoglobin (including serum proteins). The immune response parameters of immunoglobulin M and lysozyme activity were not significantly different (*p* > 0.05). Lysozyme is involved in the innate immune response of fish [69]. This enzyme functions as an antimicrobial agent that stimulates an opsonin of the complement system and phagocytic cells [70] and breaks bacterial cell walls, thereby damaging and destroying bacterial cell [71,72].

In addition, the immune system has many substances related to pathogen defense processes—especially IgM and lysozyme, which release many free radicals and cause oxidative stress. Oxidative stress causes many defects in the animal body via lipid peroxidation, DNA damage, and protein damage, consequently impairing the balance between the formation of free radicals and the body’s antioxidant capacity [73]. Fish are susceptible to attacks of reactive oxygen and have developed defense mechanisms against non-enzymatic and enzymatic antioxidative stresses that are particularly innate immune responses [74]. These systems capture and inhibit the formation of free radicals and chelate ion metals that catalyze free radical reactions. The trace minerals that are components of antioxidative enzymes include selenium, zinc, and chromium. 

Zinc is part of the group of antioxidative enzymes that are known as superoxide dismutase enzymes (Zn-SOD). Such enzymes catalyze superoxide anions and dismutate them into hydrogen peroxide and oxygen. Selenium, as a selenocysteine, is a component of the active site of glutathione peroxidase. The main function of glutathione peroxidase is the neutralization of hydrogen peroxide and organic peroxide into water and oxygen [75]. Glutathione peroxidase is a vital antioxidative enzyme that defends cells and cell membranes from oxidative damage by disrupting hydrogen peroxide and hydroperoxides via glutathione (GSH) [76,77,78]. To minimize oxidative stress in fish, the use of a feed additive supplemented with essential trace minerals can improve growth performance and enhance their antioxidant capacity [74].

In the present study, an antioxidative enzyme of SOD showed no significant effects (*p* > 0.05), whereas serum glutathione (which comprised GSSG and GSH in both treatments T1 and T2) seemed to have significant benefits. Higher glutathione levels are related to Se-dependent glutathione peroxidase activity [79]. Glutathione is present in cells, mainly in the reduced form (90% to 95% of total glutathione). The oxidation of glutathione leads to the formation of glutathione disulfide (GSSG). Intracellular GSH status appears to be a sensitive indicator of the overall health of a cell and its ability to resist toxic challenge. High levels of GSH in a cell may indicate pathological changes [80]. There have been many reviews of the relative proportion of selenium and vitamin E that is required to prevent lipid peroxidation and oxidative stress in fish [79,81,82,83,84]. A study of optimum selenium requirements for juvenile Nile tilapia showed that approximately 570 ppb of selenomethionine impacted growth and antioxidant capabilities [19]. However, in the current study, the T1 treatment with chromium-L-methionine combined with a zinc amino acid complex led to higher values of superoxide dismutase activity than the other studied groups. Sallam et al. [85] found that the dietary supplementation of zinc–methionine at 30 ppm significantly improved catalase, superoxide dismutase, glutathione peroxidase, and lysozyme activities in marbled spinefoot rabbitfish (*Siganus rivulatus*). Zinc shows the most significant benefits for the immune response as a cofactor of superoxide dismutase, and this antioxidative enzyme is essential in the defense against radical reactions and in decreasing superoxide oxygen species levels in aquatic animals [86,87,88], because optimal zinc supplementation in diets tends to increase the level of superoxide dismutase. The major function of dietary chromium is to enhance insulin efficiency by promoting blood glucose uptake in animal metabolism [89] and improving glucose utilization [90]. Moreover, a previous report demonstrated different benefits in immune response; for example, high chromium dietary yeast supplementation for rainbow trout enhanced serum lysozyme activity, innate immune response, phagocytosis ability, and respiratory bursts elicited by macrophages [15]. In the present study, the numerical values of lysozyme activity in the T1 and T2 treatments with organic mineral combinations tended to increase more than the numerical values of lysozyme activity in the control group. An appropriate supplementation of a dietary zinc amino acid complex, chromium-L-methionine, and selenomethionine can provide benefits for the immune response via antioxidative enzymes.

Safety evaluation is a crucial consideration in trace mineral supplementation. Dietary trace element additions with higher levels of zinc were shown to decrease hematocrit and hemoglobin levels [91]. Asad et al. [92] investigated the effects of diet supplementation for *Labeo rohita* and observed maximum genotoxic effects with inorganic chromium at a level of 300 ppb, in comparison with the minimum genotoxic effects observed in control diets and organic chromium treatments. In addition, Bell et al. [81,82,83] reported that hematocrit levels were reduced when selenium was absent in feed, but there were no significant effects on the level of hematological parameters in trout that were fed different levels of selenium (660, 660, and 1140 ppb). Several studies reported on the toxicity of high trace mineral accumulation that adversely impacts macrophage activation and phagocytosis activity, and which is harmful to humans and animals by decreasing oxygen production [93,94]. Therefore, we investigated the optimal levels of trace minerals in fish diets.

Glutathione peroxidase is an important enzyme that protects tissues against oxidative damage and, in this study, showed improvements that were similar to the results presented by Naiel et al. [95]. Dietary organic selenium at 360 ppb to 390 ppb as the optimal inclusion level, in the diets of Nile tilapia reared under sub-optimal temperatures, enhanced growth performance, serum biochemical indices, antioxidative enzymes, and immune responses. Selenium is an essential element in nutrition; it is the major component of selenoprotein, which is important for antioxidative stress mechanisms, thyroid hormone metabolism, and immune responses [96]. Ning et al. [97] showed that the dietary supplementation of selenium at 570 ppb could enhance weight gain and increase hepatopancreatic and serum selenium levels in juvenile Nile tilapia, depending on glutathione peroxidase activities. Accordingly, this supplementation rate is recommended for juvenile Nile tilapia. However, a study of the use of dietary selenium in rainbow trout (*Oncorhynchus mykiss*) did not demonstrate any effect on survival and growth performance in the range of 500 ppb to 800 ppb, but did demonstrate improvements in selenoprotein expression in the range of 500 ppb to 4000 ppb [98]. Following these studies, the European Union legislated that the total selenium level of feed must not exceed 500 ppb (dry mass) (Commission Regulation EU No 432/2012) [99].

## 4. Conclusions

In this study, the dietary supplementation of the organic trace minerals of chromium-L-methionine at 500 ppb, in combination with a zinc amino acid complex at 60 ppm, showed beneficial effects for red tilapia in terms of growth performance, feed utilization, and intestinal morphology. Combinations of chromium-L-methionine and a zinc amino acid complex or selenomethionine did not impact the effect of the sublethal concentration and the lethal toxicity dosage on immune-hematological responses. Under normal conditions, combining chromium-L-methionine at 500 ppb with selenomethionine at 300 ppb demonstrated improved levels of antioxidative enzymes, especially glutathione (both GSSG and GSH), which enhanced the innate immune response of red tilapia.

## Figures and Tables

**Figure 1 animals-12-02182-f001:**
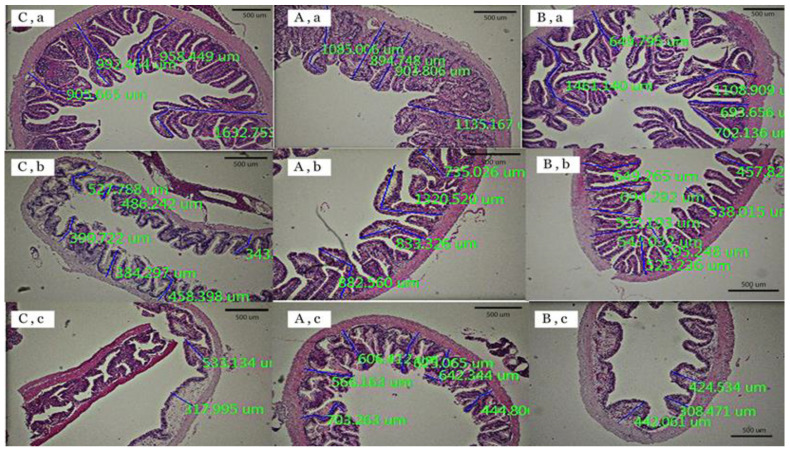
Histology slides of the anterior part (**a**), the middle part (**b**), and the posterior part (**c**) of the intestines of red tilapia that were fed with different organic trace mineral supplementations: control (**C**), T1 (**A**), and T2 (**B**).

**Table 1 animals-12-02182-t001:** The formulations of red tilapia diets.

Raw Materials (%)	Control	T1 (Zn-Cr)	T2 (Se-Cr)
Fishmeal (60%)	5.00	5.00	5.00
Corn gluten meal	8.00	8.00	8.00
Poultry meal (65% CP)	7.00	7.00	7.00
Rape seed	7.00	7.00	7.00
Fish hydrolysate	4.00	4.00	4.00
Soybean meal, dehulled	25.50	25.50	25.50
Wheat bran	8.00	8.00	8.00
Corn meal	4.00	4.00	4.00
Tapioca pulp	4.00	4.00	4.00
Tapioca chip	25.00	25.00	25.00
Lysine HCl	0.43	0.43	0.43
DL-methionine	0.17	0.17	0.17
Choline chloride	0.30	0.30	0.30
Mono-dicalcium phosphate	0.50	0.50	0.50
Salt	0.10	0.10	0.10
Micronutrient preservation premix	1.00	1.00	1.00
Zinc (zinc amino acid complex; 12%)	0.00	0.05	0.00
Chromium (Cr-L-methionine; 1000 ppb)	0.00	0.05	0.05
Selenium (selenomethionine; 1000 ppb)	0.00	0.00	0.03

Note: The micronutrient preservation premix had the following components: vitamin and mineral mix (supplements per kg of the mixed feed) comprising vitamin A, 8000 IU; vitamin D3, 1500 IU; vitamin E, 100 mg; vitamin B1, 10 mg; vitamin B2, 40 mg; vitamin B6, 40 mg; vitamin B12, 20 mcg; vitamin K3, 13 mg; ascorbic acid, 150 mg; niacin, 80 mg; pantothenic acid, 40 mg; folic acid, 4 mg; biotin, 0.5 mg; Fe, 30 mg; Mn, 25 mg; Zn, 40 mg; I, 1 mg; Cu, 5 mg; Se, 0.25 mg; and Co, 0.05 mg.

**Table 2 animals-12-02182-t002:** Growth performance of red tilapia that were fed diets with different trace minerals for 8 weeks (mean ± SD).

Growth Parameters	Control	T1 (Zn-Cr)	T2 (Se-Cr)
Final weight (g/fish) Weight gain (g/fish) ADG (g/fish)	547.72 ± 13.66 ^a^	586.58 ± 22.91 ^b^	569.49 ± 13.63 ^a^
	355.15 ± 8.60 ^a^	394.07 ± 25.91 ^b^	377.19 ± 18.71 ^a^
	6.34 ± 0.15 ^a^	7.04 ± 0.46 ^b^	6.74 ± 0.33 ^a^
SGR (%/day) FCR	1.87 ± 0.05	1.99 ± 0.11	1.94 ± 0.11
	1.55 ± 0.04 ^b^	1.36 ± 0.10 ^a^	1.44 ± 0.04 ^b^

Remarks: Data were expressed as mean ± SD and means in the same row with different letters (^a,b^) were significantly different from each other (*p* < 0.05). ADG: average daily gain; SGR: specific growth rate; FCR: feed conversion ratio.

**Table 3 animals-12-02182-t003:** Intestinal villus height of red tilapia that were fed diets with different trace minerals for 8 weeks (mean ± SD).

Intestinal Villus Height (μm)	Control	T1 (Zn-Cr)	T2 (Se-Cr)
Anterior part (foregut)	1197.89 ± 45.02 ^b^	1104.33 ± 36.98 ^b^	926.14 ± 77.38 ^a^
Middle part (midgut)	501.84 ± 104.95 ^a^	830.96 ± 88.57 ^b^	474.96 ± 42.93 ^a^
Posterior part (hindgut)	372.57 ± 22.50 ^a^	522.40 ± 57.40 ^b^	321.45 ± 72.09 ^a^

Remarks: Data were expressed as mean ± SD and means in the same row with different letters (^a,b^) are significantly different from each other (*p* < 0.05).

**Table 4 animals-12-02182-t004:** Hematological parameters of red tilapia that were fed diets with different trace minerals for 8 weeks (mean ± SD).

Hematological Parameters	Control	T1 (Zn-Cr)	T2 (Se-Cr)
RBC (106 cell/mL)	0.80 ± 0.23	0.84 ± 0.19	0.83 ± 0.20
WBC (105 cell/mL)	0.13 ± 0.05	0.16 ± 0.09	0.15 ± 0.00
Hct (%)	28.50 ± 3.70	33.75 ± 9.98	34.25 ± 4.19
Hb (g/dL)	6.39 ± 0.14	6.95 ± 0.86	6.52 ± 0.84
Serum protein (mg/dL)	6.25 ± 0.44	8.11 ± 1.14	6.80 ± 1.12
IgM (g/L)	0.49 ± 0.04	0.62 ± 0.10	0.54 ± 0.10
Lysozyme activity (units/mL)	138.75 ± 14.38	159.63 ± 25.32	163.00 ± 41.36
Superoxide dismutase (units/mL)	4.09 ± 1.31	4.72 ± 0.35	3.96 ± 0.25
Glutathione (nmol)	6.71 ± 0.33	8.42 ± 0.53	9.65 ± 2.21

Remarks: Data were expressed as mean ± SD and values in the same row were not significant (*p* > 0.05). RBC: red blood cell; WBC: white blood cell; Hct: hematocrit; Hb: hemoglobin; IgM: immunoglobulin M.

## Data Availability

All data generated or analyzed during this study are included in this article.

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
