# Peer review of "Effects of Chromium-L-Methionine in Combination with a Zinc Amino Acid Complex or Selenomethionine on Growth Performance, Intestinal Morphology, and Antioxidative Enzymes in Red Tilapia Oreochromis spp."

_animals, 2022, doi:10.3390/ani12172182_

Round 1
Reviewer 1 Report
In general, this work contain numerous shortness, concerning also methodology. I found a lot of grammar, stylish and syntax errors that make a manuscript difficult to read. The help of a native speaker is highly advised.
Line 13 – please explain what CRD stands for?Line 26 – please check the correctness of the sentence. Maybe - “The levels of antioxidative enzymes superoxide dismutase (SOD), and lysozyme activity, were not significantly different when compared to controls.” ???
Line 101 – the aim of the study is badly written and not informative enough. For example the study animal as well as brief organization of the experiment are not mentioned.
Line 106 – check English
Line 112 – Was control diet checked for the presence of Cr, Zn and Se ?
Line 159 – what were criteria for distinctions of foregut, midgut and hindgut (please refer to the literature)? Additionally, according to the recent view such division is confusing and more appropriate is “anterior,- mid,- and posterior intestine”.
Line 157, line 163 and more – Taking into account the above there is only one intestine (not Intestines !!!).
Line 166 – sections could be stained according to HE method
Line 166 – what “standard histological techniques” means ??? It is not informative enough. For example both paraffin sections and cryostat sections are widely and routinely used in morphological sciences (authors did not mention what protocol was applied).
Figure 1 - scale bars are missing
Line 24, line 312 “hindgut” and “hind gut” (line 260, 294), Which one is correct? By the way “middle” should be replaced with “mid”.
Table 2– what (ab) stands for ? Mayne more precise would be (a,b)?
Line 194 – please explain what is WST-1, please provide a source of it (code, manufacturer etc.)
Line 363 – GSSG was used for the first time in line 363 while this abbreviation was explained already in line 367
Line 363 – GSH has not been even explained.
Line 378 – there is no need to abbreviate once again SOD
Author Response
Dear Sir/Madam
First of all, thank you professor for review my manuscript and suggestion. I would like to apologize with my English gramma and make a manuscript difficult to read. However, I decided to apply English editing services and wait for few days to receive my manuscript. If they send improved manuscript to me, I will send to you as soon as possible.
For your question, I has already to revise with your suggestion in file attachment.
For the answer of your question,
Line 13 CRD = Completely Randomized Designed
Line 26 = I has already correct the word which your suggestion.
Line 101 = Please see the attachment in below of the introduction.
Line 106 = I has already correct the word which your suggestion.
Line 112 = Please see the attachment in Meterials and methods.
Line 24, 312, 159 = I has already correct the word which your suggestion in terms of anterior, middle and posterior intestine.
Line 157, 163 = I has already correct the word which your suggestion.
Line 166 = I has already revise at Line 199-200
Figure 1 = The scale bars are already show in the figure.
Table2 = Already correct the precise just only with a and b .
Line 194 = WST-1 is highly water- soluble tetrazolium salt and chemical form is [2-(4-Iodophenyl)- 3-(4-nitrophenyl)-5-(2,4-disulfophenyl)- 2H- tetrazolium, monosodium salt] which produces a water-soluble formazan dye upon reduction with a superoxide anion.
Line 363 = I has already revise at Line 88-91
Line 378 = I has already correct the word which your suggestion.
Kind Regards
Rawiwan Limwachirakhom

Reviewer 2 Report
Chromium-L-Methionine Combination With Zinc Amino Acid Complex or Sele-2 nomethionine on Growth Performance, Intestinal Morphology and Antioxydative 3 Enzymes in Red Tilapia Oreochromis spp.
This study has been made to look for the benefits of three diets supplemental with different minerals on the growth performance, intestinal morphology, and also some hematological parameters and immune response activities of juvenile red tilapia.
The methods used are well presented and they are appropriate to find posible interesting results of the fish subjected to the different experimental diets. The study finds showed that red tilapia fed diet T1 shows stadistically significant differences in the growth performance parameters studied but also in the measurement of villus height. Other practical results can also be obtained when fish are fed with the T2 diet. The results are practical to improve some aspects of the yield of the culture of the red tilapia. However, there are many studies carried out in a similar way that affect the improvement of the culture of this species. I invite the authors to read the available literature related to the objective of this study. From this perspective, with the results obtained, this study is not particularly innovative, nor does it allow increasing general knowledge in the improvement of mineral supplements for tilapia farming. In addition, the results obtained regarding hematological parameters and inmune response of experimental fish do not offer any relevant results using diets T1 and T2 compared to the diet received by the control group.
The study includes a very complete discussion, but extends too far with respect to the results achieved in the immune response of experimental fish. In fact, discussion is extends to some results that have not been obtained in the present study.
The conclusions shown do not correspond to the results obtained in this manuscript, especially regarding inmmune response of fish.
According to the arguments described above to reject this manuscript is recommended.
Author Response
Dear Sir/Madam
First of all, thank you professor for review my manuscript and suggestion. I would like to apologize with my English gramma and make a manuscript difficult to read. However, I decided to apply English editing services and wait for few days to receive my manuscript. If they send the improved manuscript to me, I will send to you as soon as possible.
In your suggestion about the innovative, I just revise my manuscript with the other parameter with the immune and stress related gene expression parameter. To improvement of mineral supplements for red tilapia or Nile tilapia farming, it should provide the benefit either growth performance and immune response among the global crisis such as climate (increasing high temperature with decrease immune or health affect growth) and finally decrease profit to farmer especially management system leads the overcrowding and affect stress condition.
In terms of other your suggestion, I have already revise and add the information as your recommendation in new manuscript. Please see the attachment.
Kind regards
Rawiwan Limwachirakhom

Round 2
Reviewer 1 Report
Most of my comments were corrected and taken into account.
Unfortunately, I still find some annoying language errors which must be correct.
For example:
Line 191 – “intestine was” instead of “were”
Line 195 – “separately collected” seems to be superfluous
Line 198 – Tissues sections can not be cut. Tissue is usually cut into sections. So, it should be “tissue samples were cut …
Line 199 – “after the fixation” instead “after fixed tissues”
And many many more
Author Response
Dear Professor,
Thank you for reviewing and commenting my manuscript. First, I would like to apologize for my language errors, which have now been rectified in my revised manuscript. If you have any questions following this revision, please contact me and I will attempt to respond as quickly as possible.
Kind Regards
Rawiwan Limwachirakhom
Reviewer 2 Report
CHROMIUM-L-METHIONINE COMBINATION WITH ZINC AMINO ACID COMPLEX OR SELENOMETHIONINE ON GROWTH PERFORMANCE, INTESTINAL MORPHOLOGY, ANTIOXIDATIVE ENZYMES, IMMUNE AND STRESS-RELATED GENE EXPRESSION IN RED TILAPIA Oreochromis spp.
(second version of the manuscript)
The current draft of this manuscript has incorporated other aspects of the research with a view to finding some results related to different experimental diets received by red tilapia, and the possible benefits obtained in relation to some productive parameters, but also with immune and stress related gene expression. This latest incorporation in the experimental design of this work seems to be intended to show more beneficial results obtained from the diets being compared.
However, from my point of view, the changes introduced do not improve the quality obtained in the previous draft, but rather add other issues and more doubts that make it more difficult to interpret and assess if beneficial results are really produced with the objectives set with this research.
In summary, the comparison of T1 and T2 diets seem to show some benefits in the growth performance parameters studied and in the measurement of villus height for red tilapia, but they offer nothing or very little regarding the expression of immune and stress related genes in the spleen of fish.
Authors might ask different issues to improve the results:
Are the concentration of Zinc Amino Acid Complex or Selenomethionine tested sufficient to obtain differences in the experimental fish?
Regarding immune response: try different times to sample the fish, try with different tissues such as head kidney, in addition of the spleen. Might a larger sample size of the fish be needed to look for differences?
Finally, I recommend a more specialized journal in the field of fish production or farming to communicate the results.
Author Response
Dear Professor,
Thank you for reviewing and commenting my manuscript.
I have provided a point-by-point answer which your comments with the attached file, please kindly see the attachment.
Kind Regards
Rawiwan Limwachirakhom

Round 3
Reviewer 2 Report
Effects of Chromium-L-Methionine in Combination with Zinc Amino Acid Complex or Selenomethionine on Growth Performance, Intestinal Morphology, Antioxidative Enzymes, and Immune and Stress-Related Gene Expression in Red Tilapia Oreochromis spp.
(third version of the manuscript)
The current manuscript that is submitted for review maintains practically intact the structure and content that was presented in the second version and that I clearly evaluated. Briefly, it contains different results that improve some aspects for the production of red tilapia, showing the beneficial effect of some of the diets, when diet T1 and diet T2 are compared with respect to the control group. Some beneficial results are also shown, showing improvements in the activity of some antioxidative enzymes and in intestinal morphology.
The current version has corrected many English expressions that improve the general understanding of the work.
Major changes:
But the current version of the manuscript continues to maintain some results that are very poor and do not improve the knowledge when they are intended to be disseminated with the current structure of the manuscript. These issues were already noted in my previous review and absolutely nothing has changed in the current version.
The only result of interest regarding the immune response of red tilapia in the present work is related to an increased protein 70 response that appears with one of the diets is used. However, it is very likely that there are important mistakes in the experimental design regarding immune rsponse of fish that the authors have carried out: different sampling times should have been done throughout the experiment and not a single sampling at the end of feeding with experimental diets (which is the case of this work), and samples of spleen and head kidney should have been taken, at least, to ensure changes in the immune response of the experimental fish. Stress-related genes may show extensive but short-lived changes, and the response may disappear after a few hours. In summary, I believe that this work should not maintain the results of the immune response of fish. The authors could save their results obtained with protein 70 and compare them in a study subsequently, where an improved experimental design is carried out, including a more complete sampling of the fish to ensure more results that allow an adequate comparison of the immune response of red tilapia using these experimental diets.
· Proposed new tittle: Effects of Chromium-L-Methionine in Combination with Zinc Amino Acid Complex or Selenomethionine on Growth Performance, Intestinal Morphology, Antioxidative Enzymes in Red Tilapia Oreochromis spp.
· Point 3.4 of the results (Expression of Immune and Stress-Related Genes in the Spleen of Red Tilapia) must be removed. The same must be done in the material and methods
· Conclusions and abstract should also be changed according with previous point.
· Lines 462 and 468: the reference Auepaiboon et al. (58) should be removed from the manuscript. It is an authors´s communication in a conference and has not passed the necessary peer review for a high impact journal.
Author Response
Dear Professor,
Thank you for reviewing and commenting on my manuscript once again. In the present edition, I have already altered some knowledge to improve the experimental results. If you have any questions or suggestions following this revision, please contact me and I will attempt to respond as quickly as possible.
Your suggestion for a new title has now been rectified and removed from my revised manuscript of the results (Expression of Immune and Stress-Related Genes in the Spleen of Red Tilapia), including material and methods, abstract and conclusion. The reference of Auepaiboon et al. (58) has been removed from the current manuscript and the references from other authors have been edited to relate and support the results in the revised version.
Please kindly find attached my resubmitted manuscript and if you have any questions of recommendations, I will try my best to follow your suggestions.
Kind Regards
Rawiwan Limwachirakhom
